# Nebulized fusion inhibitory peptide protects cynomolgus macaques from measles virus infection

Olivier Reynard [1], Claudia Gonzalez[1], Claire Dumont [1], Mathieu Iampietro [1], Marion Ferren [1], Sandrine Le Guellec[2], Lajoie Laurie [3], Cyrille Mathieu [1], Gabrielle Carpentier[4], Georges Roseau[4], Francesca T. Bovier[5], Yun Zhu [5,6], Deborah Le Pennec[7], Jérome Montharu[4], Amin Addetia[8], Alexander L. Greninger[8], Christopher A. Alabi [9], Elise Brisebard[10], Anne Moscona [5,11,12], Laurent Vecellio[4], Matteo Porotto [5,13] & Branka Horvat [1] ✉

Measles is the most contagious airborne viral infection and the leading cause of child death among vaccine-preventable diseases. We show here that aerosolized lipopeptide fusion inhibitor, derived from heptad-repeat regions of the measles virus (MeV) fusion protein, blocks respiratory MeV infection in a non-human primate model, the cynomolgus macaque. We use a custom-designed mesh nebulizer to ensure efficient aerosol delivery of peptide to the respiratory tract and demonstrate the absence of adverse effects and lung pathology in macaques. The nebulized peptide efficiently prevents MeV infection, resulting in the complete absence of MeV RNA, MeV-infected cells, and MeV-specific humoral responses in treated animals. This strategy provides an additional means to fight against respiratory infection in non-vaccinated people, that can be readily translated to human trials. It presents a proof-of-concept for the aerosol delivery of fusion inhibitory peptides to protect against measles and other airborne viruses, including SARS-CoV-2, in case of high-risk exposure.

Measles virus (MeV), a member of the *Paramyxoviridae* family of single-stranded negative sense RNA viruses, is one of the most infectious microorganisms worldwide, with a primary reproduction rate of 12–18[1]. Despite the availability of a safe and effective vaccine, measles causes 3 to 4 million cases annually, claimed 207.500 lives in 2019, and remains a leading cause of childhood death from vaccine-preventable diseases in many developing countries[2]. Although incidence has decreased considerably from 2000 to 2016 (from 145 to 18 per million), measles has increased since 2017[2] and is expected to further increase in incidence as a result of the SARS-CoV-2 pandemic and the intercurrent delays in childhood immunization programs and resultant "immunity gaps" in the population[3–6]. In addition, in developed countries, imported outbreaks pose a significant risk for immunocompromised people who rely on herd immunity and cannot receive the current live vaccine[7].

MeV is an airborne pathogen, transmitted by inhalation of respiratory droplets and smaller aerosol. Initial infection targets susceptible cells in the respiratory tract[8,9]. After an incubation period of 7 to 10 days, the acute phase is characterized by fever, oculo-respiratory inflammation, cough, and Koplick spots[10]. The characteristic erythematous skin rash occurs around 14 days after infection[11] when MeV infects cells in the epidermis[11,12]. MeV is amplified in regional lymphoid tissues, followed by systemic infection when MeV-infected lymphocytes and dendritic cells (DCs) migrate into the subepithelial cell layers and transmit MeV to the epithelial cells of various tissues[13]. After amplification of MeV in the epithelia, progeny virus is released into the respiratory tract, either free or as dislodged infectious centers consisting of infected cells[14]. Infection generally resolves after three weeks, but is often followed by immune suppression which may last for

several months and is responsible for numerous complications associated with measles[15,16]. Rare but severe complications of measles involve central nervous system infection progressing to lethal MeV encephalitis[7].

MeV uses several receptors to infect target cells. The virus binds to human CD150 (SLAMF1, hSLAM) to infect macrophages, DCs, and lymphocytes[17,18], and the attachment is aided by the CD209 (DC-SIGN) co-receptor[19]. Alternatively MeV can use nectin-4 as a receptor[20] to infect the basolateral side of the airway epithelial cells, a process that promotes viral dissemination[21]. Recent data suggest that entry may also occur from the apical side of the airway epithelium in a nectin-4-independent manner[8,9]. Finally, a role for CADM1 and 2 in neuroinvasion has been postulated[22]. Infection is initiated following the attachment of the hemagglutinin (H) protein to one of MeV receptors; H then activates the fusion protein (F), and the ensuing rearrangement of F promotes insertion of the hydrophobic fusion peptide into the facing cellular membrane. A second folding event occurs driven by interaction between N- and C- terminal heptad repeat (HR) regions (HRC and HRN domains respectively) of F, completing virus-cell membrane fusion[23]. Our previous work has demonstrated that peptides derived from HRC region can interfere with the second folding event required

for virus-to-cell fusion during MeV infection[24,25]. A dimerized version of a peptide corresponding to the HRC region, conjugated to a cholesterol moiety (referred to as "HRC4" peptide, Fig. 1d), inhibited the fusion process in cell culture and in organotypic brain cultures[24]. HRC4 peptide administered intranasally to cotton rats and to humanized transgenic mouse models of lethal measles disease led to reduction of the viral titer in cotton rat lungs and a significant increase in survival of mice[24,25].

Advantages of inhaled protein therapeutics include the non-invasive needle-free drug delivery route, and the ease of depositing drugs directly in the lungs while limiting systemic toxicity[26]. Since the approval of inhaled Dornase alfa for treating pulmonary disease in cystic fibrosis, several peptides have been under clinical development for inhaled delivery[27]. Nebulizers can be used for high dose delivery with limited drug formulation development[26–28]. In the present study, a mesh nebulizer was used to deliver MeV fusion inhibitory peptides to nonhuman primates (NHPs) - cynomolgus macaques - a well-characterized model that recapitulates measles infection in humans[29]. The prototype applied in this study uses a piezo-electric generator to push the drug solution through a microperforated metal sieve, allowing a fast and silent drug delivery[28]. Small diameters of the

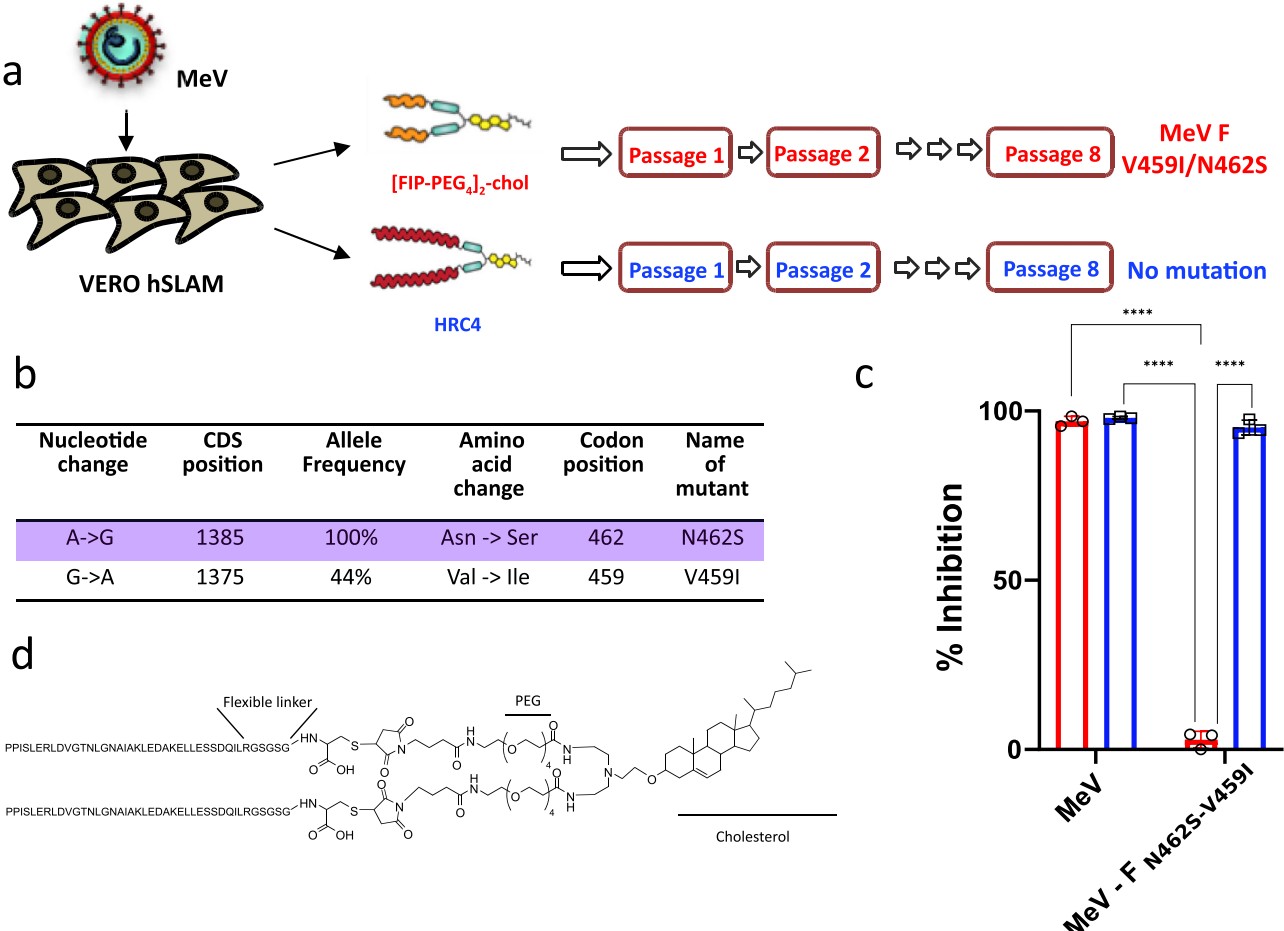

**Fig. 1 | HRC4 lipopeptide treatment does not generate MeV escape variants.**
**a** Schematic presentation of viral passaging. Measles virus MeV IC323-eGFP was serially passaged 8 times on the Vero-hSLAM cells in presence of either 1 μM [FIP-PEG4]2-chol (red) or HRC4 peptide (blue), added to the culture after the infection. Virus was titrated after each passage and 100 Plaque Forming Units (PFU) used for each subsequent infection. Sequencing of viral RNA after the 8th passage revealed two mutations in the F-HRC domain for the infection done in the presence of [FIP-PEG4]2-chol: V459I and N462S, while no mutations in the presence of HRC4 peptide were found. **b** The most frequent mutation events in F after MeV IC323 was serially

passaged 8 times on the Vero-hSLAM cells in presence of [FIP-PEG4]2-chol.
**c** Inhibition of cell-cell fusion mediated by MeV F bearing the indicated mutations by either 5 μM [FIP-PEG4]2-chol or HRC4 peptide, using HEK-293T cells transfected with hSLAM and the omega reporter subunit of β-gal, incubated with cells co-expressing viral glycoproteins (IC323 H and F) and the alpha reporter subunit of β-gal (**** $p < 0,0001$, Two-Way ANOVA multiple comparison test, $n = 3$ biologically independent samples in 2 independent experiments). Data are presented as mean values ±SD. Source data are provided as a Source Data file. **d** Schematic of the HRC4 lipopeptide, used in the further study.

sieve pores generate aerosols smaller than 5 μm, enabling efficient pulmonary drug delivery[27]. Using this mesh nebulizer for respiratory administration of fusion inhibitory peptide we effectively inhibited MeV infection in the macaque model. These results open novel perspectives for antiviral prevention strategy against measles and possibly other airborne viruses, including SARS-CoV-2.

## Results

### Treatment with HRC4 lipopeptide does not promote selection of drug-resistant variants

Generation of escape variants is a concern with any antiviral[30], we thus initially tested for emergence of peptide resistant MeV variants in cell culture. Recombinant MeV IC323-eGFP[31] was grown on Vero-hSLAM cells in the presence of either 1 μM HRC4 peptide or another fusion inhibitory peptide (FIP) carbobenzoxy-(Z)-D-Phe-L-Phe-Gly peptide[32], -[FIP-PEG$_4$]$_2$-Chol-, which was dimerized and coupled to cholesterol like HRC4[33]. Viruses were sequenced after eight passages (Fig. 1a). In the [FIP-PEG$_4$]$_2$-Chol-treated cells, two mutations in MeV HRC domain were identified, in the same residues as described previously under the selective pressure of the unconjugated FIP, $V_{459}$ and $N_{462}$[34] (Fig. 1a, b). However, no mutations were identified in the HRC4-treated MeV. The FIP resistant MeV variants were susceptible to inhibition by HRC4 peptide, as determined using a quantitative fusion assay (Fig. 1c). Fusion between cells expressing wt or variant MeV glycoproteins and cells expressing hSLAM was measured by β-galactosidase complementation in the presence of 5 μM of FIP or HRC4. FIP inhibited membrane fusion mediated by the wild-type F but did not affect the fusion mediated by F-$N_{462}$S or F- $N_{462}$S/$V_{459}$I mutated proteins. In contrast, HRC4 inhibited membrane fusion mediated by both the wild-type and mutant F proteins (Fig. 1c). These data demonstrated the absence of HRC4-resistant MeV mutants following multiple viral passages and strengthen the selection of HRC4 lipopeptide (Fig. 1d) for further preclinical development of MeV fusion inhibitory peptide.

### Analysis of the dose and schedule of HRC4 administration in murine model of MeV infection

We next evaluated the dose and schedule of HRC4 peptide by intranasal administration in the humanized murine transgenic model of CD150xIFNα/βR KO mice, previously shown to be very susceptible to MeV intranasal infection[35,36]. Mice were treated with HRC4 peptide (1 or 0.1 mg/kg) either twenty-four or six hours prior to intranasal infection with a lethal dose of MeV-IC323-eGFP (10$^4$ PFU). All mock-treated mice succumbed to the infection by day 12 post-infection (p.i.), while the treatment with the HRC4 (1 mg/kg) six and twenty-four hours before infection led to a significantly higher survival rate (p < 0.0001, Mantel-Cox test) (Fig. S1).

Based on these results and previous work[24,25], we selected for further studies a 4 mg/kg dose, given 3 times by nebulization, 24 h and 6 h before infection, and 24 h after infection, to optimize the antiviral effect of HRC4 in primates which are highly susceptible to MeV, considering the possible loss of peptide delivered through nebulization. This choice was also driven by a recently published study of the therapeutic three-time nebulization of antiviral compound in respiratory syncytial virus (RSV)-infected children[37].

### Characterization of an aerosol device for lipopeptide delivery into the lung alveoli

MeV infection of the respiratory tract targets the lung alveoli[8], we therefore engineered an inhaled strategy and used a customized mesh nebulizer to deliver the HRC4 lipopeptide aerosol deep into the respiratory tract, to block virus infection. The particle size measurement of aerosol generated following the nebulization of either HRC4 peptide or saline solution (0.9% NaCl) using a prototype mesh nebulizer with a 3 μm pore sieve and a prototype face mask (Fig. 2a) was assessed by laser diffraction (Fig. 2b). The nebulizer devices delivered

particles with an average size of 4 μm, in terms of the Volume Mean Diameter (VMD), of both peptide and saline solution, at a flow rate of 0.32 – 0.46 ml/min (Fig. 2b, detailed in Table S1). Approximately 58% of particles were smaller than 5 μm, which is the aerosol size that reaches the airways[27], where MeV infection initiates.

The inhibitory effects of HRC lipopeptide in vitro were evaluated before and after nebulization to address the possibility that nebulization itself could inducing aggregation or degradation of peptide with resultant loss of activity[26,27] (Fig. 2c, d). Nebulization of HRC4 did not cause any loss of activity. Cytotoxicity of the nebulized HRC4 before and after nebulization was evaluated in vitro using Vero-E6 cells (Fig. 2d). No measurable cytotoxicity was observed at doses ranging from 0.5 nM to 4 μM, indicating that the HRC4 therapeutic index is higher than 500 (4 μM / 8 nM)[38].

### Biodistribution and safety of nebulized HRC4 in cynomolgus macaques

Delivery of aerosol by the customized nebulizer to the macaques was measured by scintigraphy imaging of animals nebulized with $^{99m}$TC-DTPA-labeled in NaCl 0.9% solution, chosen since HRC4 peptide solution and NaCl shared similar aerodynamic properties (Table S1). After nebulization, 40% of the total aerosolized product reached the respiratory tract, with 11.4% distributed into the lungs (Fig. 3a). The deposition of the HRC4 peptide in the respiratory tract was further analyzed using anti-HRC4 antibodies for the immunofluorescent detection of the peptides in lungs of macaques, sampled either immediately after nebulization (15 min) or 16 h and 24 h later (Fig. 3b and Fig. S2). As expected from the scintigraphy imaging, analysis of all three lung regions revealed the presence of HRC4 within the alveoli surface area, suggesting peptide distribution throughout the lungs following the nebulization.

Search for a production of IgE in the serum of nebulized animals by ELISA did not reveal presence of any HRC4-specific IgE, nor increase in IgE (Fig. S4a, b), up to 4 weeks after nebulization. Histological analysis of lungs collected from macaques nebulized with either saline or HRC4 did not show any abnormalities specific to HRC4 group (Fig. S3 and S4c), but highlighted a minimal although noticeable eosinophil infiltration in all animals that developed MeV infection, in accord to previous reports of MeV infection in rhesus macaques[39]. In addition, in support to the absence of allergic reactions to peptide treatment, the histological analysis of lung sections did not present any signs of hyperplasia of mucus cells and smooth muscles, characteristic for allergic reactions of the respiratory tract.

We further analyzed whether peptide could reach the blood circulation following the nebulization. HRC4 was found in low concentration (below 1 nM) in the serum of nebulized animals, up to 96 h after a third nebulization (Fig. 3c), while it was absent in the urine. Low entry of the peptide into the circulation did not lead to the active immunization of animals, as HRC4 specific antibodies were not found in the serum 28 days after nebulization (Fig. 3d). In addition, no adverse effects (pyrexia, allergic reaction) were observed during the 28 days after peptide nebulization. Histological analysis of lungs collected from nebulized animals did not reveal any abnormalities (Fig. S3, S4c).

Biochemical parameters in the plasma and cellular composition of the blood were evaluated immediately before treatment and 1, 2, 3, 6, and 28 days after nebulization of either saline or HRC4 peptide in non-infected and MeV-infected NHP, to search for early and late toxic effects of the aerosol delivery (Fig. 4). Analysis of the numerous hematological and biochemical parameters in noninfected animals were within the physiological levels[40] and did not show significant variations between groups. In both control and peptide group, few animals experienced transient increase of creatine kinase above physiological values[40] (Fig. 4a), which can be linked to intramuscular anesthesia, as described previously[41]. In addition, flow cytometry

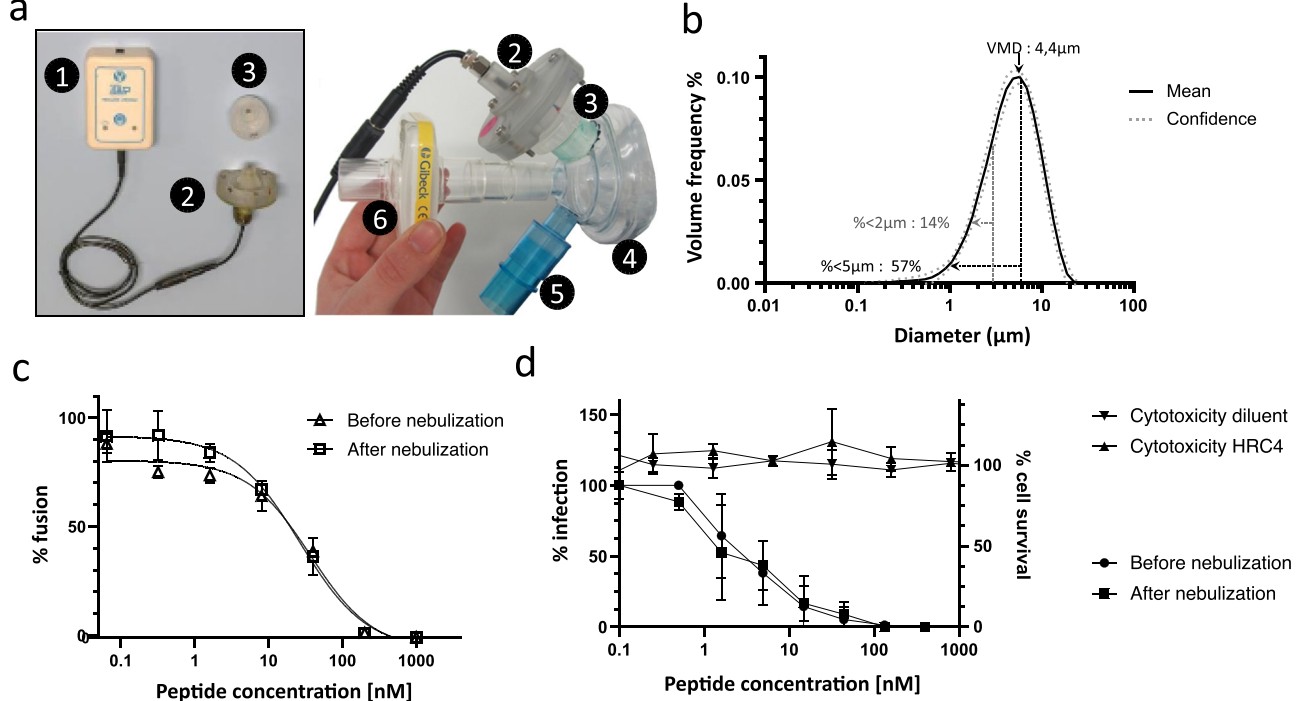

**Fig. 2 | Utilization of the prototype mesh nebulizer with low Volume Medium Diameter preserves functional activity of HRC4 lipopeptide. a** Composition of the prototype mesh nebulizer used in experiments: 1. electronic controller; 2. piezo-electric vibrator; 3. reservoir containing the mesh; 4. facemask; 5. one-way inspiratory valve; 6. absolute filter. **b** Graphical representation of particle size distribution obtained from laser diffraction analysis of aerosolized 3 ml of HRC4 lipopeptide. Plain line presents the mean values of four different nebulizers used in the study and dotted line standard deviation. Percentage of particles <5 µm and <2 µm present the fraction of aerosol below the indicated size, corresponding to the aerosol penetrating into either lung in general (<5 µm) or into alveolar regions of lungs (<2 µm). Volume Mean Diameter (VMD) presents the mean size of generated aerosols. **c** Fusion inhibitory activity of HRC4, measured before or after peptide nebulization, using β-gal complementation assay. Data represent the mean ± SD of two independent experiments. **d** Antiviral activity of HRC4 measured prior and after nebulization, determined by $IC_{50}$ measurement using plaque reduction assay on Vero-hSLAM cells, and cytotoxicity assay, performed by assessing of cell viability 96 h after addition of HRC4 by MTT assay. Data represent the mean ± SD of of three independent experiments (Man-Whitney test, $p = 0,8535$). Source data are provided as a Source Data file.

monitoring of the composition of major PBMC populations in blood did not reveal any significant changes in animals nebulized with peptides compared to values obtained before the nebulization (Fig. S6).

## Modelling of the virus and peptide deposition on the lung surface area

To estimate the coverage of the lung surface area of a NHP following the aerosolization of a peptide present within the nebulised particles in the relationship to the administered viral inoculum, schematically presented in the Fig. 5, we next performed the mathematical modelling. The estimation is based on the calculation of either virus or peptide dose per unit of lung surface as described in the FDA guidance for inhalation product[42]. The calculation took into consideration the number of droplets formed following the administration, the distribution of viral particles and the amount of peptide molecules covering the pulmonary area[43]. Calculation of the number of infectious viral particles (Nv) administered in the macaque's lung took into account the method of viral administration which consists in delivering 10000 plaque-forming unit (PFU) of virus, (Nv) in a form of liquid (Vv), through the endo-tracheal tube, leading to the formation of a thin liquid layer in the lung conductive airways[44]. Taking into account the worst-case scenario, ie a liquid thickness (Tv) of 7 µm[45] recovering the epithelia in the conductive airways, we calculated the surface (Sv) covered by the 5 ml liquid volume (2 ml inoculum and 3 ml washout):

$$Sv = Vv/Tv = 7140\ cm^2 \qquad (1)$$

Then, the calculation of the virus concentration (Cv) in the region of infected surface lung was performed using the following formula:

$$Cv = Nv/Sv = 1.4\ PFU/cm^2 \qquad (2)$$

Determination of the number of droplets administrated in the macaque lung was based on results presented in the Fig. 2a, showing that the deposition fraction in the lung (E) is around 10%, with the nebulizer charge Vp = 3 ml, giving thus 0.3 ml in a form of deposited droplets in the lung. Knowing the mean diameter of droplets (Dp = 4 µm), we can calculate the number of deposited droplets (Nd) as follows:

$$particles\ volume = \frac{4\pi}{3}\left(\frac{Dp}{2}\right)^3 \qquad (3)$$

$$Nd = \frac{E\ Vp}{particles\ volume} = \frac{6E\ Vp}{\pi\ Dp^3} = 9 \times 10^9\ droplets \qquad (4)$$

Based on the peptide concentration (400 nmol/ml) and the nebulizer charge (Vp), we can calculate the number of peptide (Np) deposited in the lung as follow:

$$Np = Na\ Cm\ Vp\ E = 7 \times 10^{16}\ peptide\ molecules\ (Na : Avogadro\ number) \qquad (5)$$

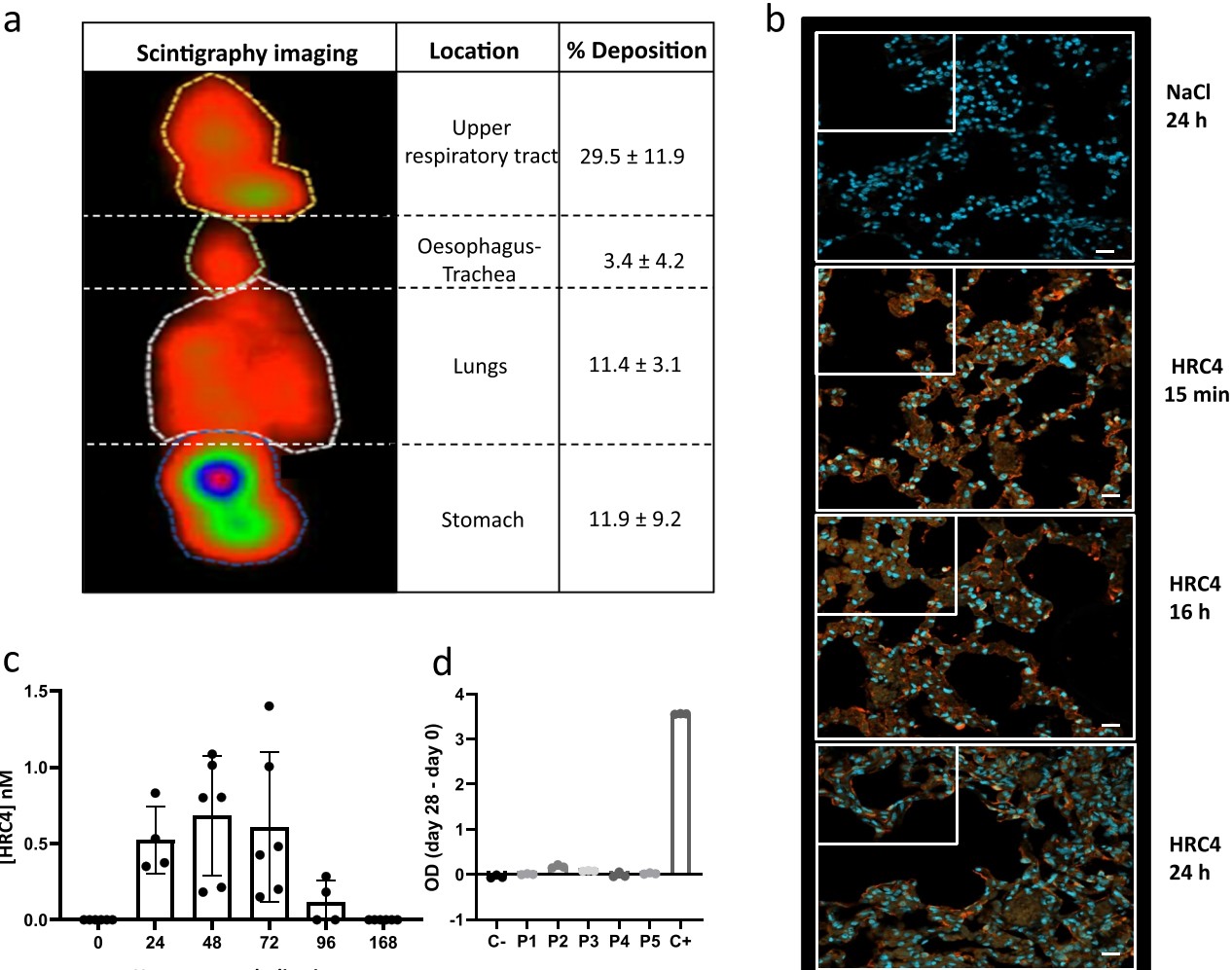

**Fig. 3 | Delivery of the aerosol into macaques' lungs. a** Representative scintigraphy imaging of different regions of interest in a cynomolgus monkeys nebulized with $^{99m}$TC-DTPA tracer (74 Mbq) in 3 ml NaCl 0.9%, using a prototype mesh nebulizer and measured by E-cam gamma camera. Mean values ±SD of the distribution of aerosol deposition were determined from the digitalized images obtained in four experiments. **b** Lung localization of the aerosolized HRC4 peptide analyzed in cranial lobe lung sections from monkeys nebulized under mechanical ventilation for the indicated time: 15 min ($n=1$), 24 h ($n=2$) or 16 h ($n=1$), prior to euthanasia. Staining was performed with rabbit anti-HRC peptide and goat anti-rabbit Alexa 555 (orange staining) and DAPI was used to stain nuclei (blue staining)

(Scale bar: 20 μm). **c** Quantification of HRC4 in the NHPs' serum after peptide nebulization, by ELISA. Histograms present mean values ± SD from 2 independent experiments. Significant increase of HRC4 in the serum, following the nebulization, was confirmed using One way ANOVA (Prism 8.4.3 software, $p=0.0004$).
**d** Determination of the presence of anti-HRC4 antibodies (IgG, IgA and IgM) by ELISA in the serum of peptide-nebulized macaques, from either HRC4 biodistribution or MeV-infection study, P1-5 (values obtained at day 0 are subtracted from those at D28 for each individual animal). Negative control (C-) corresponds to serum of a naïve macaque and positive control (C +) to rabbit anti-HRC4 antiserum. Source data are provided as a Source Data file.

If we consider a homogenous deposition of the liquid in the lungs, we can calculate the concentration in terms of number of peptide molecules per surface of lung as the ratio of the number of peptide molecules (Np) and the macaque's lung surface:

$$[R] = \frac{Np}{\text{alveoli area} \times \text{total number of alveoli}} = \frac{7.22 \times 10^{16}}{(0.19\,\text{mm}^2 \times (2 \times 57.8 \times 10^7))} \quad (6)$$
$$= 3 \times 10^{11}\,\text{peptide}/\text{cm}^2$$

Consequently, in the lung of the macaque, a surface of 7140 cm$^2$ was covered with the virus 1.4 PFU / cm$^2$. On the same lung surface, we deposited by nebulisation $3 \times 10^{11}$ peptide/cm$^2$. This concentration of peptide was administrated homogeneously to the totality of the lung. Interestingly, we have obtained a $2 \times 10^{11}$ ratio between the peptide and the virus per cm$^2$ in the infected lung surface, being thus largely in favor of the peptide deposition (Fig. 5), and highly encouraging for the further in vivo assay using MeV infected NHP.

Finally, when the estimation of the peptide and virus deposition in the in vitro tissue culture was performed using a similar type of calculation, the large excess of the peptide to virus was also obtained ($5 \times 10^{11}$ peptides/cm$^2$, Fig. S5), in accord to the highly efficient inhibition of virus infection seen in vitro (Fig. 2c, d).

### Nebulized HRC4 peptide protects cynomolgus macaques from MeV infection

To assess antiviral efficacy of nebulized HRC4 peptide in NHPs, groups of 3 animals were infected with 10$^4$ PFU MeV IC323-eGFP by intra-tracheal inoculation, and either mock-treated with nebulized saline solution (C1, C2, and C3) or treated with nebulized HRC4 peptide (P1, P2, and P3) twenty-four and six hours before infection and twenty-four hours after infection (Fig. 6a). The NHPs were housed in cages accommodating two animals per cage for ethical reasons, so that one cage contained one peptide-treated and one saline-nebulized macaque. The animals were monitored for 28 days for the appearance of

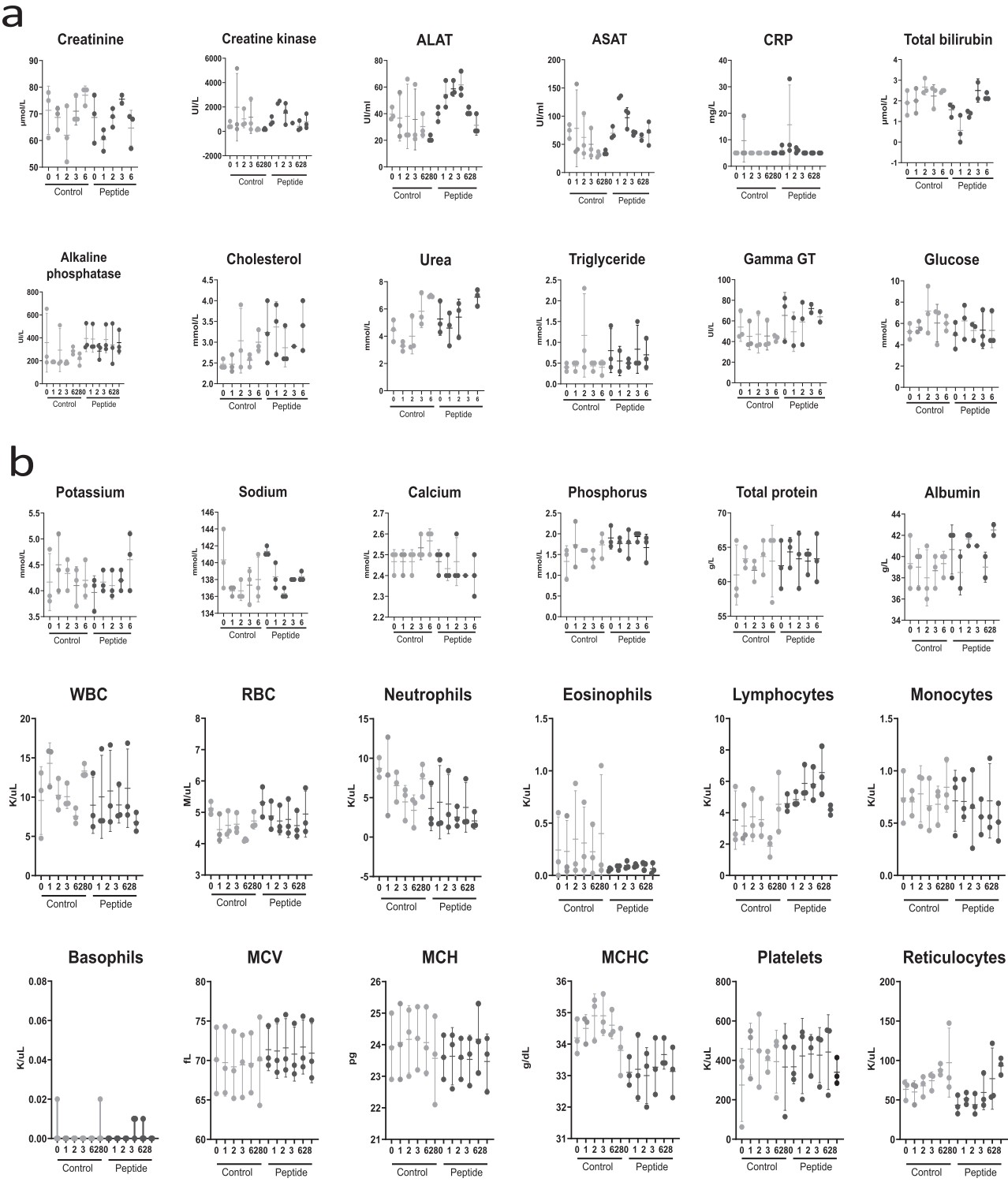

**Fig. 4 | Evolution of biochemical and haematological parameters in the blood cynomolgus monkeys following the nebulization of either saline (control) or HRC4 (peptide). a** Concentration of indicated biochemical parameters in blood of monkeys, measured on days 0, 1, 2, 3, 6 and 28 after nebulization of either saline solution (0.9% NaCl, $n = 3$) or HRC4 peptide (4 mg/kg, $n = 5$); **b** Haematological parameters measured at days 0, 1, 2, 3, 6 and 28 after nebulisation of either saline solution (0.9% NaCl, $n = 3$) or HRC4 peptide (4 mg/kg, n = 5). ALAT Alanine aminotransferase, ASAT Aspartate aminotransferase, CRP C-reactive protein, MCV Mean corpuscular volume, WBC white blood cells, RBC red blood cells, MCH Mean corpuscular, MCHC Mean corpuscular haemoglobin concentration. Data are presented as mean ± SD, with individual points corresponding to each analysed animal. Source data are provided as a Source Data file.

clinical signs, including temperature, weight, and behavior changes, none of which were observed during the experiment.

As in human infection, measles infection in NHPs induces a skin rash that can be followed macroscopically during infection with eGFP-encoding MeV[17]. Observation of the fluorescent skin rash is facilitated by utilizing a blue LED light, particularly in cynomolgus macaques where the red skin rash is less apparent than in rhesus macaques[46]. After infection with MeV-IC323-eGFP, skin and oral mucosa were monitored under blue light every three days postinfection (p.i.). All mock-treated animals, but no HRC4-treated animals, had a GFP-

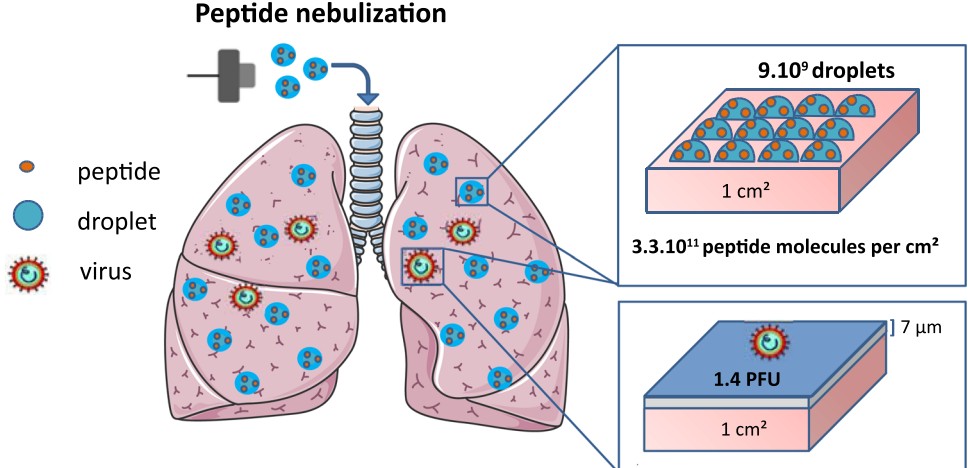

**Fig. 5 | Schematic presentation of the peptide and virus deposition in the lungs.** The calculation based on the estimation of the peptide dose per unit of lung internal surface, taking into consideration the number of droplets formed following the administration and the peptide molecules into the pulmonary area. Following the nebulization, 11% of the formed droplets reached the lungs (as shown in Fig. 3a), representing a density of $9\times10^9$ droplets/cm² of lung internal surface and containing $3\times10^{11}$ peptide molecules/cm². Instillation of the viral inoculum ($10^4$ PFU in 5 ml) via endotracheal tube leads to the virus dispersion in the lung conductive airways in the form of 7 μm thin liquid layer (presented in blue color, covering the maximum surface of 7140 cm²). This represents 3.5 % of the total lung surface area, giving the density of infectious particles of 1.4 PFU per cm² of airways and estimated ratio between the peptide and the virus is $2\times10^{11}$ per cm² of the lung surface.

fluorescent rash (Fig. 6b, c). The fluorescent rash appeared as early as day 6 p.i. (animal C3) and lasted until day 16 (C1 and C2) or up to day 28 p.i. (C3), mainly located in the mouth (tongue, palate, gum, and chin) and skin (preferentially on armpits, groin, and back) (Fig. 6b, c).

MeV infects and replicates in circulating immune cells[13,15]. Infection of PBMCs was detected by flow cytometry on day 3 p.i. in animals C1 and C3, on day 6 p.i. in animal C2 and at a low level in animal P2 at day 9 p.i. (Fig. 6d, left panel). Consistent with eGFP expression, viral RNA followed a similar kinetic trajectory (Fig. 6d, central panel). Viremia lasted 9 to 16 days in mock-treated animals and 6 days in HRC4-treated animal P2. This NHP was housed in the same cage with mock-treated C3, which exhibited a high level of MeV infection and most probably transmitted it as a secondary infection to P2 after the end of the peptide treatment, confirming the high contagiousness of MeV infection. The viremia of animal C3 peaked at day 6 p.i. while animals C1 and C2 peaked at day nine p.i. However, despite the delayed infection, animal P2 had lower level of viremia than mock-treated animals. Oral shedding of virus was monitored by viral genome quantification in RNA extracts from throat swabs (Fig. 6d, right panel). Low levels of viral RNA were found at early time points and likely represent leftovers of the initial inoculum, while viral shedding peaked at day 9 p.i. and lasted up to day 16 p.i. in saline-treated animals. Consistent with the PBMC results, viral RNA was only detected late and transiently at day 13-16 p.i. in the swabs from animal P2.

### HRC4 nebulization prevents MeV infection in peripheral immune cells

Transient immunosuppression associated with leukopenia is a hallmark of MeV infection in humans[15] and is observed in MeV-infected NHPs. Hematological monitoring of MeV-infected NHPs demonstrated a transient leukopenia in the saline-treated group and at later time points in the infected animal P2, as observed in previous reports[17,47], with total white blood cell and lymphocyte counts decreased on day 6 following MeV infection in saline-treated animals, but remained stable in HRC4-treated macaques (Fig. 4b and S6). Leukopenia lasted longer in saline-treated macaques (days 6-16 p.i.) compared to animal P2 (days 9-13 p.i.). In infected saline-treated animals, leukopenia was associated with lymphopenia, which was not observed in the fully protected macaques P1 and P3 (Fig. S6).

Flow cytometry studies revealed a transient decrease in B cells (CD20+) between day 9-13 p.i. in mock-treated MeV-infected animals (Fig. S7). The proportion of CD3+CD4+ and CD3+CD8+ T cells remained unchanged despite a decrease in absolute lymphocyte number (Fig. S7). Evaluation of MeV-infected cell phenotype showed only a few CD14+ monocytes positive for GFP between day 6 and 13 p.i, with CD4+ T lymphocytes and CD20+ B lymphocytes constituting the main targets of the virus (Fig. 7a). Of the total cells infected, 40-60% were T cells and 20% were B cells, with a peak of infection day 6 (C3) or day 9 (C1, C2 and P2) p.i. The magnitude of infection of CD3+CD8+ T cells and CD14+ monocytes was lower (Fig. 7a) and the majority of infected cells among PBMCs were CD4+ lymphocytes (Fig. 7b; data are displayed for animals that had detectable and sufficient numbers of infected cells.). Interestingly, peptide-treated animal P2, who likely was infected later by transmission from its non-treated cage-mate, had a very low percentage of all infected cell populations, ranging between 5-30 times lower than saline-treated animals, suggesting an anti-viral effect of HRC4 nebulization followed by secondary infection from the co-housed actively infected C3 macaque (Fig. 7a).

### Humoral immune response in animals combatting MeV infection

MeV infection induces life-long immunity to reinfection, characterized by the generation of a MeV-specific lymphocyte response[15]. We evaluated peripheral blood B cell phenotype and serum antibody responses in MeV-infected macaques using flow cytometry, to track the presence of unswitched (CD20+ CD27+ CD38+ IgD+) memory B cells, secreting only IgM, and class-switched (CD20+ CD27+ CD38+ IgD-) memory B cells, known to secrete IgG, IgA or IgE (Fig. 8a). Both B cell populations increased from day 3-6 p.i. and day 9 p.i, respectively, in mock-treated animals C2 and C3, although the response of C1 was much lower. The secondarily infected HRC4-treated animal P2 displayed a similar but delayed increase in both class-unswitched and -switched B cell populations. Notably, both P1 and P3 HRC4-treated animals were fully protected against MeV, and B cell populations remained stable without any noticeable increase.

MeV-specificity of the B cell response was further confirmed by serological analysis (Fig. 8b, c). All saline-treated animals seroconverted after MeV infection, with a high MeV antibody titer on day 28 p.i. The secondarily infected animal P2 had a slightly lower total

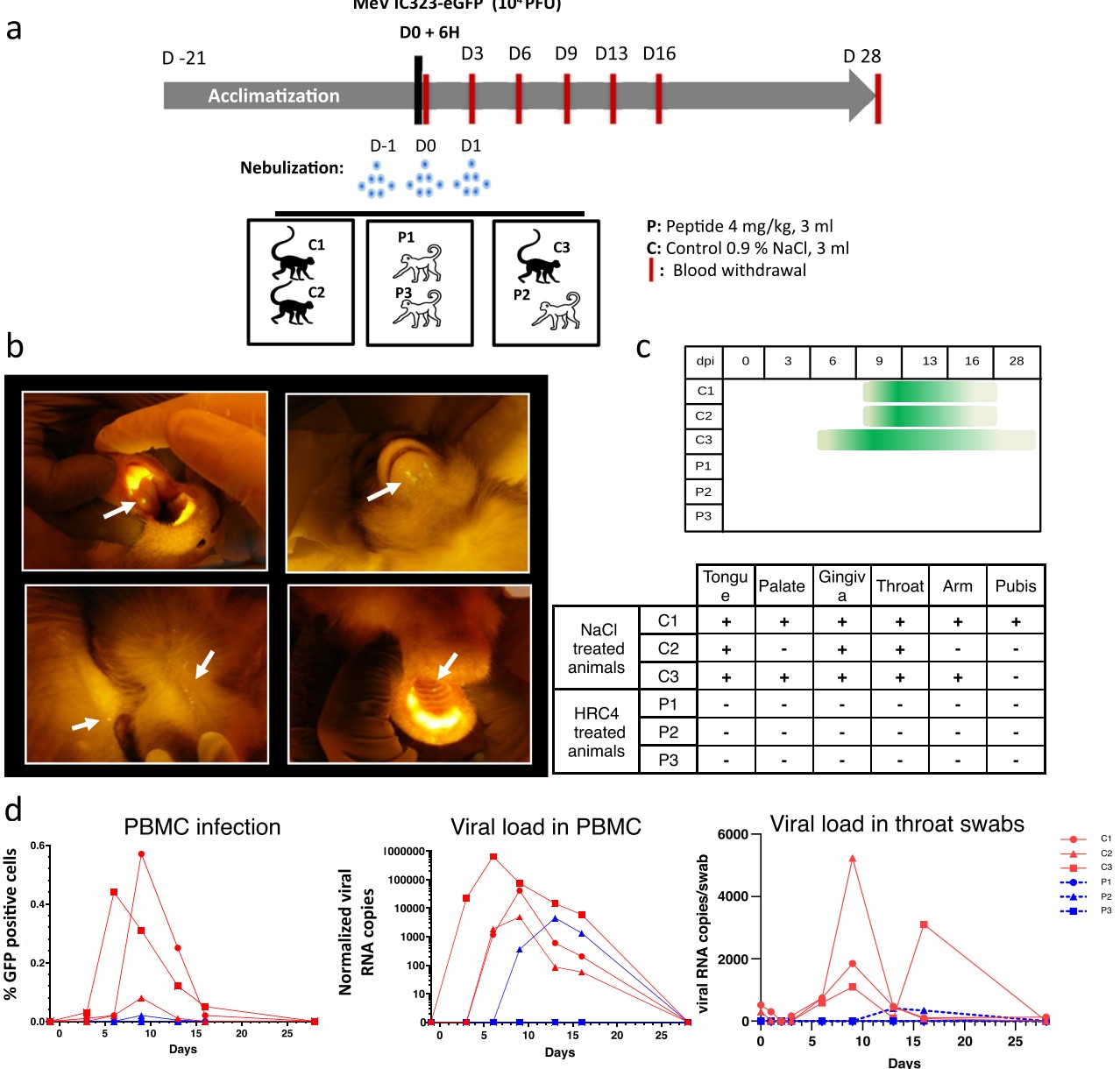

**Fig. 6 | HRC4 nebulization protects monkeys from clinical manifestation of MeV infection. a** Experimental design: cynomolgus monkeys (3 animals/group) were either nebulized with 3 ml of NaCl 0.9% (control group, C) or with 3 ml HRC4 peptide 4 mg/ml (experimental group, P), 24 h and 6 h before and 24 h after intra-tracheal infection with $10^4$ PFU MeV IC323-eGFP. Blood samples were taken every 3 days for the first 16 days and fluorescence of the skin and mucosa tested at that time points. **b** Macroscopic manifestation of MeV infection, typical fluorescent rash observed on tongue, skin (back and chin) and palate (marked with white arrows), monitored under anesthesia using a blue light with orange filter. **c** Duration of the clinical signs in MeV-infected animals followed daily (dpi: days post infection). **d** Analysis of viremia by quantification of the percentage of GFP+ peripheral blood mononuclear cells (PBMC) in the blood by flow cytometry and MeV-specific RNA in PBMCs and in throat swabs by RT-qPCR, during the course of infection. Source data are provided as a Source Data file.

antibody titer (Fig. 8b). All seropositive animals secreted neutralizing antibodies with $SN_{50}$ values ranging between 546 and 3465 (Fig. 8c). The absence of seroconversion of HRC4-treated animals P1 and P3 correlated with the lack of viral replication and the distinct composition of lymphoid blood compartment in those animals, underlining the efficient and robust protection provided by the nebulized HRC4 lipopeptides.

## Discussion

Airborne infection is transmitted through small aerosolized particles suspended in the air and is responsible for spreading many important infectious diseases of humans and animals. In this study, we pioneered a nebulization approach to inhibit highly contagious MeV infection in the NHP model with fusion inhibitory peptides. As measles continues to present a significant health problem worldwide[2], there is a need for prevention modalities in addition to vaccination for those who either cannot be vaccinated or do not respond appropriately to vaccination. In the current study, we adopted an approach based on immunovirological and technological research, to develop a drug and device that can be adapted to treat human patients. Fusion inhibitory HRC4 peptide provided complete protection to MeV challenge after delivery by nebulization. This needle-free therapy may find acceptance among people when compared to other routes of administration[48–50]. The production of 4 μm aerosolized particles by the device used in this study supports its use for targeting MeV and possibly other airborne viruses.

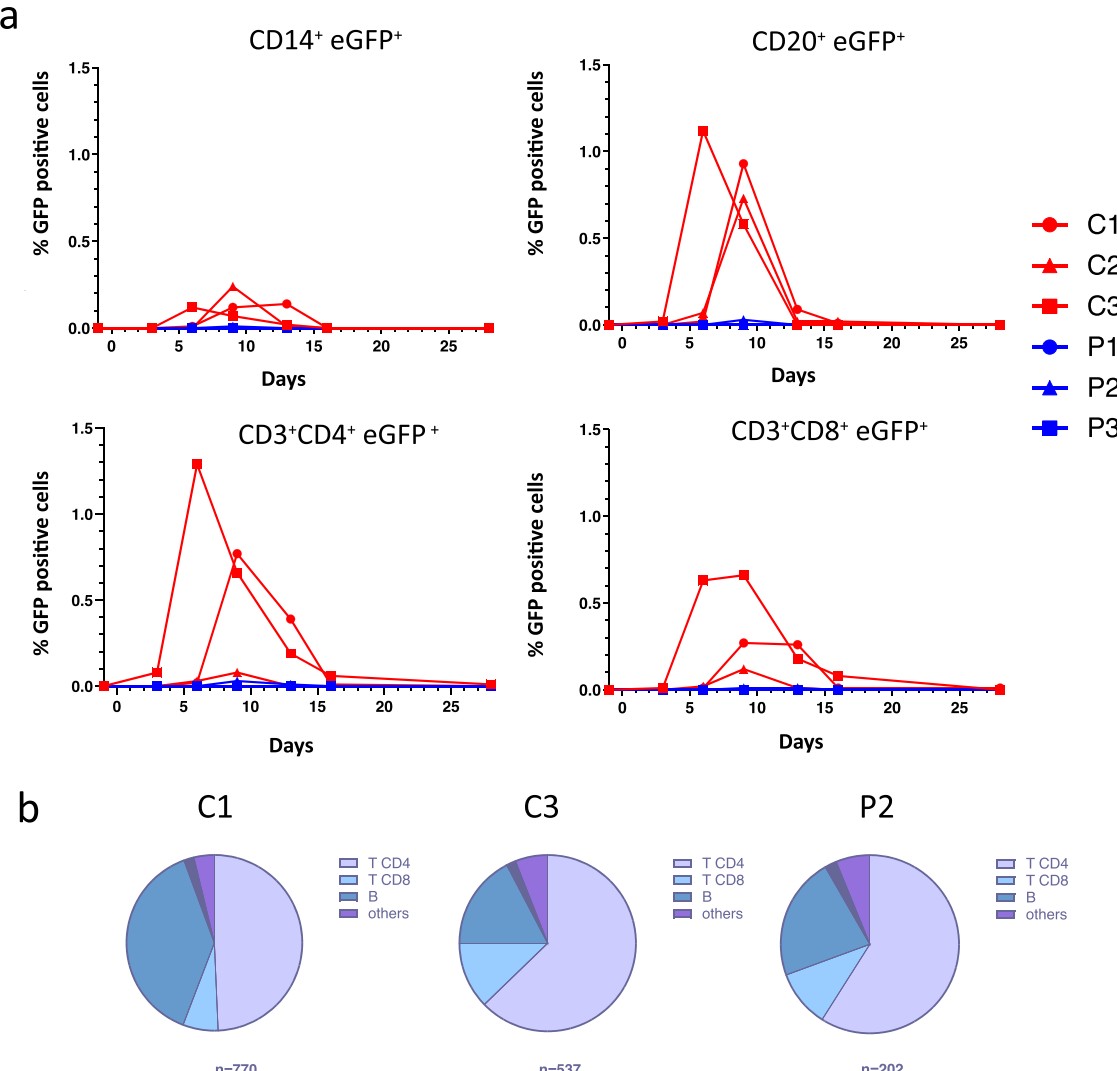

**Fig. 7 | Nebulization of HRC4 peptide protects PBMCs from MeV infection.**
**a** Quantification of MeV eGFP positive cells in indicated PBMC subpopulations by flow cytometry: CD14+ monocytes, CD4+, CD8+ and CD20+ lymphocytes in MeV-infected cynomolgus monkeys by flow cytometry, following the nebulization of either 0.9% NaCl (C) or HRC4 peptide (P). CD4+ T lymphocytes were characterized as CD3+CD8-, and CD8+ T lymphocytes were characterized as CD3+/CD8+; B-lymphocytes were characterized as CD3-/CD20+ cells. **b** Analysis of the contribution of each lymphocyte subpopulation among infected PBMCs; results are presented as the percentage of each analyzed cell population among the infected cells on the day of peak of MeV infection (day 6 for C3 and day 9 for C1 and P2; C2 is not displayed due to a low number of infected cells). Numbers below the graphs correspond to the number of analyzed cells for each presented animal. Data were acquired on a MACSQuant® 10 flow cytometer (Miltenyi). Source data are provided as a Source Data file.

MeV infection of cynomolgus macaque mimics both pediatric respiratory infection physiology and mild MeV infection in humans[47,51–55]. Our results in this model have demonstrated that aerosol peptide administration using the prototype mesh nebulizer device results in efficient deposition of HRC4 peptide into lungs and persistence of detectable peptide twenty-four hours after the nebulization. This application of HRC4 peptide aerosol represents a promising initial step that supports its use in humans, where this device should work even better in the absence of the anatomical constraints of primates[56]. In addition, the excellent safety profile, absence of any adverse reaction, and non-immunogenic character of the compound following nebulized administration support the strategy for human use. Finally, in contrast to several other antiviral compounds[30,34,57], repeated passage of MeV in the presence of HRC4 lipopeptide did not elicit viral escape mutants (Fig. 1), suggesting that frequent administration may not promote development of drug-resistant variants.

HRC4 peptide treatment abrogated the development of MeV infection in two out of three animals, measured by the absence of fluorescent rash, PBMC infection, viremia, viral shedding, and MeV-specific immune response. The third HRC4-treated animal (P2), housed with the mock-treated highly-infected animal C3, developed a late paucisymptomatic infection without rash, delayed and reduced viremia, low shedding, and late immune activation of B lymphocytes. Thus, animal P2 might have been protected from MeV initial challenge by HRC4 aerosolization and acquired a MeV infection from animal C3 once the activity of the nebulized HRC4 decreased. This hypothesis agrees with the mode of action of HRC4 fusion inhibitor peptide, which is expected to prevent initial infection through daily administration and provide protection that is estimated to endure at least twenty-four hours. Although the approach presented in this work was not tested in humans, we applied both in vitro and in vivo models and mathematical modeling of the peptide and the virus deposition in the lungs to support future development and to predict how the drug delivery system will behave in humans.

Animals that experienced MeV infection developed a transient leukopenia consisting mainly of lymphopenia and moderate

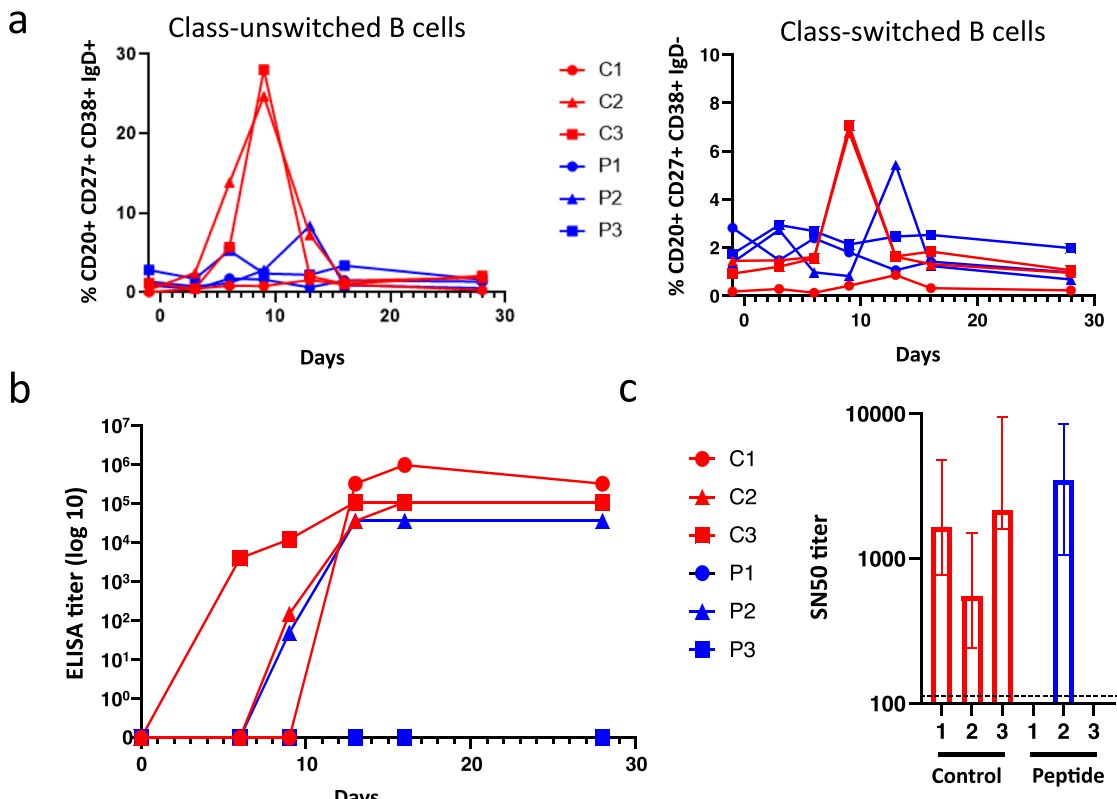

**Fig. 8 | Establishment of the humoral immune response in animals that develop MeV infection. a** Analysis of the presence of class-unswithched IgM secreting B cells (CD20$^+$ CD27$^+$ CD38$^+$ IgD$^+$) and switched memory B cells secreting IgG, IgA or IgE (CD20$^+$ CD27$^+$ CD38$^+$ IgD) in the peripheral blood of NHPs, in following the nebulization of either 0,9% NaCl (C) or HRC4 peptide (P) and MeV infection; **b** Quantification of total MeV-specific immunoglobulin by ELISA; plotted values present the reciprocal values of last serum dilution with detectable optical density measure. **c** Sero-neutralization assay performed using plaque reduction test, following the infection of Vero-hSLAM cells with MeV IC323-eGFP (50 PFU/well). SN$_{50}$ values were calculated by regression using Prism 8.3 software (Nonlinear fitting, variable slope, R$^2$: 0.87-0.98); histogram bars represent SN50 values, dashed line represents detection limit and error bars represent confidence interval derived from slope fitting, variable slope, R$^2$: 0.87-0.98, $n = 2$, biologically independent dilution series in 2 independent experiments. Source data are provided as a Source Data file.

monocytosis. Those parameters were in accordance with previous descriptions of MeV tropism and disease course in the cynomolgus macaque[47,58]. The recorded infection levels were within the range of those observed by De Vries et al.[47] and slightly below those surveyed by De Swart et al.[17]. MeV-infected cells were preferentially CD3$^+$CD4$^+$ T cells, followed by CD20$^+$ B cells, CD3$^+$CD8$^+$ T cells, and subsequently CD14$^+$ monocytes, as previously described[47]. One animal (C3) experienced a faster acute infection with a detectable viremia on day 3 p.i. that peaked on day 6 p.i. and lasted until day 16 p.i., while in other animals, viremia peaked at day 9.

Seroconversion was observed in all animals that developed MeV infection, as evidenced by the appearance of neutralizing antibodies and by the activation of B lymphocyte populations. In animals C2 and C3, class-unswitched memory B cells increased at 3-6 days p.i., while class-switched memory B cells started to appear day 9 p.i., consistent with primary IgM production followed by a switch in immunoglobulin class leading to a secondary secretion of IgG, A or E. Unexpectedly, animal C1 only displayed a minor and delayed modulation of its memory B cell populations, while both total immunoglobulins and neutralizing antibodies were produced. However, animal C1 demonstrated an unusual distribution of lymphocyte populations with B cell counts, representing 35% of PBMCs compared to 6% on average for the other animals. In line with this discrepancy, total Ig was delayed compared to the two other animals from the same group. HRC4-treated animals P1 and P3 developed neither signs of seroconversion nor immune cell activation, underlining the profound protection provided by HRC4 peptides. Despite displaying limited and delayed B

cell activation following the paucisymptomatic infection, animal P2 seroconverted.

The last four years have witnessed a drastic increase in measles cases despite a highly effective vaccine[3–6], suggesting the importance of developing an additional safe prophylactic strategy to support global MeV eradication. The approach developed here, nebulization of fusion inhibitory peptides, should be clinically applicable. A fusion inhibitory peptide inhibitor of HIV entry (enfuvirtide) has been commercialized to treat HIV-infected patients by subcutaneous administration[56], and an oral fusion inhibitor for respiratory syncytial virus (prestatovir) is in clinical trials[59]. Our efforts over the last decade have been directed to design such an entry inhibitor approach for MeV[24,25,60–62]. The results presented here show that nebulization of our entry inhibitor peptides significantly reduces the clinical impact of MeV infection in NHP, providing a proof-of-concept for antiviral prophylaxis to be developed for humans. This strategy holds potential for protecting immunocompromised people who rely on herd immunity and cannot receive the current live MeV vaccine, since nebulized peptide is capable to completely halt viral infection. In addition, certain specific conditions (blood transfusion, pregnancy, transplantation, tuberculosis, etc) may require a postponement of measles vaccination[63], presenting one of the situations where HRC4 nebulization may provide a solution until vaccination is again possible and could even be continued after vaccination, until appearance of protective immunity.

Protection against one of the most contagious aerosol-transmissible viral diseases[1] is a critical achievement, suggesting the

potential of the nebulization approach for airborne enveloped viruses with similar entry pathways including SARS-CoV-2[64,65] or highly pathogenic Nipah virus[66]. In the case of viral evolution or the emergence of a new strain, the rapid development of a new antiviral based on a modified peptide sequence is feasible. Efficacy of nebulization as an administration route suggests that these antivirals are practical, possible, and within reach for use in the field where outbreaks occur. In parallel to vaccines, when available, and protective equipment, i.e. masks, aerosolized peptides may provide an additional shield to fight against extending outbreaks of airborne transmissible viruses, notably in case of high risk exposure like indoor high density people grouping (aircrafts, exhibitions, lectures ....). This antiviral strategy forms the basis for efficacious and timely emergency response immediately following identification of an emergent airborne virus which uses a similar fusion mechanism for viral entry[25,33,61,64], now with the added benefit of a suitable delivery device.

## Methods

The research presented in the manuscript complies with ethical and regulatory board of the Ethics Committee for Animal experimentation and French Ministry of High Education and Research.

### Study design

The primary objective of this study was to evaluate the biodistribution, safety, and antiviral efficacy of nebulized MeV fusion inhibitory lipopeptide HRC4. The initial evaluation of the peptide dose and the administration schedule was performed in CD150xIFNα/βR knock-out (KO) mice, highly susceptible to the intranasal MeV infection[35], using 52 mice (31 males and 21 females) separated into five groups. The study was completed using the NHP model of cynomolgus macaque, well-characterized to reproduce MeV infection similar to what is seen in humans[29]. In the setting of nebulization experiments with NHPs, the number of animals was minimized to 2 times two macaques for the study of biodistribution, pharmacokinetics, and toxicology, two macaques to analyze scintigraphy gamma camera imaging of the aerosol delivery and to 2 groups of 3 NHPs, nebulized with either HRC4 peptide or saline as a control, all 6 infected with MeV, for the study of antiviral efficacy of the tested lipopeptide.

### Cells and virus

Vero cells expressing human SLAM (Vero-hSLAM, ECACC 04091501) and HEK293T (ATCC CRL 3216) were grown in DMEM glutamax (Gibco) supplemented with 10% fetal bovine serum (FBS), glutamine and antibiotics (100 U/mL of penicillin and 100 μg/mL of streptomycin) in 5% $CO_2$ incubators at 37 °C and were tested negative for *Mycoplasma spp* (MycoAlerte, Lonza LT07-318). Recombinant MeV-IC323 expressing the gene encoding eGFP (MeV-IC323-eGFP, Genbank accession number: LC420351.1) was generated using reverse genetics in 293-3-46 cells as previously described[67], using the plasmid encoding MeV IC323-eGFP kindly provided by Y. Yanagi (Kyushu University, Fukuoka, Japan)[31]. Viral stocks were propagated and titrated on Vero-hSLAM cells.

Vero-hSLAM cells were infected with 100 PFU of MeV IC323-eGFP, then incubated for 2 h at 37 °C and further treated with several concentrations of peptides to promote the emergence of escape variants. Viruses were collected after five days and passaged similarly eight times. Viral sequencing was performed using metagenomic next-generation sequencing as described previously[68]. Briefly, RNA was extracted from 50 μL of culture harvest using the Quick-RNA Viral Kit (Zymo) and treated with TURBO DNase (Thermo Fisher). cDNA was generated from the DNase-treated RNA using Superscript IV Reverse Transcriptase (Thermo Fisher) and random hexamers (IDT), followed by second-strand synthesis via Sequenase Version 2.0 DNA Polymerase. The resulting double-stranded cDNA was then purified with the DNA Clean & Concentrator Kit (Zymo). Libraries were constructed

from 2 μL of cDNA using Nextera XT kit (Illumina) and sequenced on 1×192 bp Illumina MiSeq runs. Sequencing reads were adapter and quality trimmed using Trimmomatic v0.38. Variants present at an allele frequency greater than 10% and greater than 10x depth were identified with LAVA (https://github.com/greninger-lab/lava) using a previously sequenced MeV strain (NC_001498) as the reference genome. All variants were manually confirmed by mapping sequencing reads to the same MeV reference strain in Geneious v11.1.4. Sequencing reads are deposited in NCBI BioProject PRJNA828179.

### MeV infection of mice

CD150xIFNα/βR KO mice[35,36], generated by crossing hSLAM transgenic mice into an IFN Receptor α/β deficient background, were bred at the institute's animal facility (PBES, ENS-Lyon, France) as heterozygotes for hSLAM transgenes. Three to 4 weeks old mice (males and females) were infected by intranasal route (i.n.) with 10 μl of MeV IC323 in both nares ($10^4$ PFU/mouse) under isoflurane anesthesia. CD150xIFNα/βR KO mice were given i.n. either 0.1 or 1 mg/kg of HRC4 peptide 24 h or 6 h before the infection. Control mice received the same number of administrations of the diluent. All animals were observed and weighed daily for four weeks and those showing clinical signs (neurological symptoms, ataxia, lethargy) were euthanized. The protocol was reviewed by the Regional Ethical Committee CEC-CAPP and French Ministry of High Education and Research and approved under the agreement reference APAFIS N° 21141-2019042916294753v5.

### MeV infection of NHP

Cynomolgus monkeys (*Macaca fascicularis*) were obtained from Bioprim® (Baziege, France). The effect of HRC4 nebulization on MeV infection in NHPs was analyzed at the BSL2 primate facility at the University of Tours, France. The experiment received approval from the French Ministry of High Education and Research and was performed under the agreement reference MESRN N°29992-2021022209579514. Six healthy female cynomolgus macaques, weighting 2.6-4 kg, aged 2-4 years, were housed in groups of 2 animals/cage. All animals were confirmed by serology to be negative for MeV and canine distemper virus. Three macaques included in the control group (C1, C2, C3) were nebulized with 0.9% of NaCl, while the others received HRC4 nebulization (P1, P2, P3). The ethical obligation to house at least 2 NHPs together in the same cage resulted in two animals from different groups, C3 and P2, being co-housed within the same cage, increasing the risk of late MeV transmission between those two animals. The experiment started after 21 days of acclimatization with the nebulization for a 10-15 min period, using a prototype mesh nebulizer, with either 3 ml of peptide (4 mg/ml) or saline (0.9% NaCl), 24 h and 6 h before infection, and 24 h post-infection. Animals were infected under medetomidine/xylazine anesthesia by MeV IC323-eGFP with $10^4$ PFU in 2 mL by the intra-tracheal route, and macaques were followed for 28 days before euthanasia.

Blood samples for hematology analyses were collected at days −1, 0, 1, 2, 3, 6, 9, 13, 16 and 28. Blood samples for flow cytometry analyses were collected at days −1, 3, 6, 9, 13, 16 and 28. Throat swabs were collected at days 0, 1, 2, 3, 6, 9, 13, 16 and 28, using cotton swabs. Oxygen saturation and heartbeat were monitored by a Radical-7® Pulse CO-Oximeter (Masimo) and breathing was monitored by a Dräger Primus anesthesia machine. Hematological parameters were measured on a Procyte DX (IDEXX) and biochemical parameters were evaluated on a Konelab 30 (Thermo Fisher). Development of fluorescent rash was followed using a FastGene Blue/Green LED Flashlight (Nippon genetics).

### Study of biodistribution and toxicology in NHP

In the initial study, the pharmacokinetics and toxicology of HRC4 peptide nebulization were analyzed in 2 two-year-old healthy female

cynomolgus macaques, weighing 2.8 kg, at Cynbiose, Marcy l'Etoile, France, accredited by AAALAC. The protocol was approved by the Ethics Committee of VetAgro-Sup and French Ministry of High Education and Research under number 146 (MESR N° 2016072117544328). Animals were initially acclimatized to their designated housing room for two weeks and gradually trained during that period to remain calm when being held by the operators during manipulations (blood sampling, monitoring of body temperature) using a reward-based training regimen. Before nebulization, animals were anesthetized with ketamine (5 to 15 mg/kg) and midazolam (0.5 to 1.3 mg/kg) by intramuscular injection and then placed on a baby chair. Aerosols were administered through a face mask connected to the prototype mesh nebulizer (DTF-Medical, Saint-Etienne). For the pharmacokinetic phase, the aerosolized peptide was administered via the same face mask as above to anesthetized animals on day 0. Blood samples were collected 4, 8, 24, 48, and 72 h after nebulization. The animals had a washout period of 17 days, and blood samples were collected on day 21, the day preceding the start of the toxicology phase. For the toxicology phase, the nebulized peptide was administered via a face mask to anesthetized animals daily on three consecutive days (22, 23, and 24). Urine and blood samples were collected on day 25, and animals were euthanized for organ collection.

To evaluate immediate HRC4 biodistribution into lungs, one animal from the MeV-infection experiment (C1) was nebulized with 3 ml (4 mg/ml) immediately after euthanasia under mechanical ventilation (Dräger Primus anesthesia machine). For mid-term biodistribution, animals were nebulized 24 h (C2) or 16 h (C3) before euthanasia.

### Scintigraphy Gamma camera imaging of the aerosol delivery into NHP

A study of the biodistribution of the nebulized aerosol was performed at the University of Tours, France. Following European recommendations, 2 five-year-old healthy female cynomolgus macaques, 3–4 kg, were housed under conventional conditions in the animal facility. The experimental protocol was conducted according to European regulations for animal experimentation and approved by French Ministry of High Education and Research under the agreement reference MESR N° 11682#2017100217166146. Animals were acclimatized to laboratory conditions and trained to breathe an aerosol with a facemask spontaneously. Aerosol generated from 3 ml of 0.9% NaCl mixed with 74 MBq of DTPA radiolabeled with technetium 99 m ($^{99m}$Tc-DTPA) was administered through a facemask connected to the prototype mesh nebulizers, as used in MeV infection experiments (Fig. 2a). Deposition of aerosol was extrapolated based on the $^{99m}$Tc-DTPA signal measured at the end of the nebulization using a gamma camera (Orbiter 75 Ecam, Siemens healthcare, Erlangen, Germany)[69]. The nebulizer charge was measured by counting the radioactivity in the syringe (that contained $^{99m}$Tc-DTPA) before and after loading the nebulizers. Immediately after aerosol delivery, the animals were imaged using the gamma camera. The post-anterior static scintigraphy acquisition was performed for 120 s. The amount of $^{99m}$Tc-DTPA deposited into airways and stomach and remaining in the nebulizer was determined from the digitalized images taking into account the tissue attenuation coefficients, previously determined by perfusion scintigraphy (intravenous injection of $^{99m}$Tc-macroaggregates of albumin). The organ body outline was specified using a specific Region Of Interest (ROI), and the lungs were delineated using the perfusion scan ROI. The aerosol dose delivered to different organs of NHPs is reported as a percentage of the nominal dose placed in the nebulizer for that given experiment, taking into account the decay of technetium for all measurements.

### Peptide synthesis

Unconjugated MeV HRC peptide and FIP (Carbobenzoxy-(Z)-D-Phe-L-Phe-Gly peptide) were purchased from Shanghai Ruifu Chemical Co.,

Ltd. Bis-maleimide cholesterol was custom made by Charnwood Molecular, Ltd. HRC4 and FIP-dimer cholesterol were conjugated and purified as previously described[33]. For the in vivo experiments in mice, HCR4 peptide was initially dissolved in DMSO to 50 mg/ml and stored at −80 °C. Peptides were then diluted in water to reach either 0.1 mg/kg or 1 mg/kg for intranasal administration. For nebulization of macaques, HRC4 peptide, soluble in water, was dissolved in Milli-Q water filtered to obtain a final concentration of 4 mg/ml. The pH of peptide solution was adjusted to 7 and stabilized using HEPES buffer. Peptide preparations were kept at 4 °C for four days or at −80 °C for the long-term storage.

### Fusion assay

HEK 293 T cells transfected with hSLAM-coding plasmid and the omega reporter subunit of β-gal ("target cells") were incubated with cells co-expressing viral glycoproteins (IC323 H and F) and the alpha reporter subunit of β-gal ("effector cells") in the absence or presence of inhibitory peptides at the concentration of 5 μM. In the absence of peptides, fusion between the target and effector cells permits reconstitution of β-galactosidase activity, quantified using the luminescence-based kit, Galacto-Star β-galactosidase reporter gene (ThermoFisher) and analyzed on an Infinite 200PRO Life sciences luminometer (Tecan). Percent inhibition was calculated as the ratio of the relative luminescence units in the presence of a specific concentration of fusion inhibitory peptide and the relative luminescence units in the absence of inhibitor, corrected for background luminescence.

### Cell toxicity assay

Vero cells were incubated at 37 °C in the presence or absence of the indicated peptides at indicated concentrations up to 5 μM HRC4 peptide as added into the media, and the cells were incubated a 37 °C. According to the manufacturer's guidelines, the viability was determined after 24 h using the Vybrant MTT (3- (4, 5-dimethylthiazolyl-2) −2, 5-diphenyltetrazolium bromide) cell proliferation assay kit. TritonX-100 (1%) was used as a positive control. Absorbance was read at 540 nm using a Tecan M1000PRO microplate reader.

### Viral load quantification by RT-qPCR

Viral RNA was extracted using Qiamp Viral RNA Kit (Qiagen) for sera and swabs samples and Nucleospin Kit (Macherey Nagel) for PBMCs. Viral load was evaluated by one-step RT-qPCR (NEB Luna® Universal One-Step RT-qPCR kit) using MeV-N-specific primers (MeV-N FW: GTG ATC AAA GTG AGA ATG AGC and MeV-N Rev: GCT GAC CTT CGA CTG TCC T) and GAPDH primers if necessary (GAPDH FW: CACC-CACTCCTCCACCTTTGAC, GAPDH REV: GTCCACCACCCTGTTGCTG-TAG). PCR amplification was recorded on a Step One plus apparatus (Thermo). All samples were run in duplicates, and results were analyzed using the ABI StepOne software v2.1 (Applied Biosystems).

### Laser diffraction measurement

The aerodynamic performances of the aerosols generated by the prototype mesh nebulizer were determined by laser diffraction using a Spraytec™ instrument (Malvern Instruments Ltd., Malvern, UK) and the Spraytec inhalation cell (Malvern Instruments Ltd., Malvern, UK) connected to an aspiration carried out by a vacuum pump set to 30-50 L/min[70]. The prototype mesh nebulizers (n=4) were loaded with 3 ml of either NaCl 0.9% or the HCR4 peptide (4 mg/ml, dissolved as described above) and then connected to the inhalation cell. Nebulization duration was notified at the end of the complete aerosolization of the loaded 3 ml. Diffraction data and volume distribution were automatically registered by the Spraytec software. The volume mean diameter VMD, in μm, the respirable fractions inferior to 5 μm (%<5 μm) and inferior to 2 μm (%<2 μm) were calculated by the software.

The output of nebulizer was determined by the difference between the weight of the nebulizer before and after nebulization and was expressed in percentage of the loaded volume. The output rate of each nebulizer (in ml/min) was then determined as the ratio between the output and the nebulization duration. At least, the residual volume corresponding to the volume of liquid remaining in the reservoir at the end of the nebulization was also determined by weighting the nebulizer before loading it and after nebulization.

## Enzyme-linked immunosorbent assay (ELISA)

Determination of the HRC4 concentration in the serum and urine of macaques after the third nebulization was determined by ELISA. Maxisorp 96 well plates (Nunc) were coated overnight with purified rabbit anti-MeV-F HRC antibodies (Genescript) (5 μg/ml) in carbonate/bicarbonate buffer pH 9.2 at +4 °C. Plates were washed twice using PBS followed by incubation with 3% BSA in PBS (blocking buffer) for 60 min. Then, the blocking buffer was replaced with 2 dilutions of each sample in 3% PBS-BSA in duplicate and incubated for 90 min at room temperature (RT). Wells were washed 3 times using PBS, and the peptide was detected using an HRP-conjugated rabbit custom-made anti-MeV F HRC antibody (1:1500) in blocking buffer for 2 h at RT. Detection of HRP activity was measured by using the TMB substrate (Thermo Fisher) and by reading absorbance at 405 and 620 nm on Multiskan FC reader (Thermo Fisher). The standard curves were established for each peptide, using the same ELISA conditions as for the test samples and the detection limit was determined to be 0.04 nM.

Sera of MeV-infected NHPs were tested for the presence of anti-MeV antibodies by ELISA. Briefly, MeV nucleoprotein, produced as described previously[71] was coated onto 96-well ELISA plates overnight (1 μg/well). Plates were blocked using a mix of PBS 1X-Milk 5% for 30 min at room temperature. Serial dilutions (1:50 and then 1:3 serial dilutions until 1:2952450) were done in PBS 1X-Tween 0.05%-Milk and incubated 2 h at RT. Secondary antibody goat anti-monkey IgG-A-M conjugated to horseradish peroxidase (HRP) diluted 1/30 000, (Sigma-Aldrich) was incubated for 1 h at 37° and plates were revealed using TMB substrate solution (Thermo Fisher). Optical density was measured at 450 and 620 nm using an ELISA reader (Thermo Fisher) and the absorbance difference between 450 nm and 650 nm was determined and corrected for blank readings. The serum sample was considered positive when its value was higher than three times the average value obtained with the negative sera of the same dilution and the results were expressed as reciprocal value of the last serum dilution giving the measurable values.

Sera of HRC4-treated NHPs were tested for the presence of anti-HRC4 antibodies by ELISA. Briefly, HRC4 was coated in carbonate buffer pH 9.2 overnight (1 μg/well). Plates were blocked using a mix of PBS 1X-Milk 5% for 30 min at room temperature. Dilutions (1:20) were done in PBS 1X-Tween 0.05%-Milk and incubated 2 h at RT. For the detection of IgG, IgA and IgM, secondary antibody goat anti-monkey IgG-A-M conjugated to horseradish peroxidase (HRP), (Sigma-Aldrich) was incubated for 1 h at 37° and plates were revealed using TMB substrate solution (Thermo Fisher). For the detection of IgE, all reagents were diluted in 3% BSA, secondary antibody rabbit anti-monkey IgE conjugated to biotin (Gentaur) diluted 1/2000, was incubated for 1 h at 37° and then with streptavidin HRP (RnD system) diluted 1/2000; plates were revealed using TMB substrate solution (Thermo Fisher). Optical density was measured at 450 and 620 nm using an ELISA reader (Thermo Fisher) and the absorbance difference between 450 nm and 650 nm was determined and corrected for blank readings. Rabbit anti-HRC4 Ab followed with anti-rabbit Ab coupled to HRP was used as a positive control. Evaluation of total IgE in the serum was performed using the monkey IgE kit (Ozyme) following the manufacturer's recommendation.

## Sero-neutralization

Neutralizing Ab titers were determined using plaque reduction number test. Serial dilutions of sera (1:3) in DMEM medium containing 2% FCS were mixed with 50 PFU of MeV IC323-eGFP, incubated 30 min at 37 °C and layered on Vero-hSLAM cells in 6 well-plates for 90 min. The inoculates were replaced by DMEM 3% FBS / CMC 0.6% and plates were incubated for 3 days at 37 °C. Plaques were counted after crystal violet staining, and relative neutralization titers were defined as the reciprocal dilutions of sera samples that completely inhibited the cytopathic effect of MeV. Data were analyzed by Prism 8.3 software to calculate $SN_{50}$ values (non-linear regression, [inhibitor] vs. response, variable slope fitting).

## Histopathology and immunofluorescence

Lung tissue samples were collected at necropsy and fixed by immersion in 10% neutral buffered formalin (Sigma) for a minimum of 14 days. The tissue samples were trimmed, processed and embedded in paraffin. 4 to 5 μm-thick sections were cut. For histopathology evaluation, slides were routinely stained with hematoxylin-eosin (HE) or with hematoxylin-eosin-saffron (HES). For the study of the pulmonary infiltration by eosinophils, the histopathologic evaluation was performed by a board-certified veterinary pathologist.

To assess the bioavailability of the HRC4 peptide on lungs, sections were stained and imaged by confocal microscopy as described previously[24]. Briefly, after being blocked and permeabilized in 0.1% TritonX100, 5% BSA solution, slices were sequentially incubated with a rabbit anti-HRC4 (Genscript) overnight at 4 °C and with a secondary donkey anti-rabbit alexa-555 (Thermo Fisher) diluted 1/500 and DAPI (1 μg/ml) for 1 h at room temperature. Slides were imaged using a Zeiss LSM800 confocal microscope and processed by ImageJ1.52p.

For hematoxylin-eosin staining, formalin-fixed tissues were processed and embedded in paraffin and tissue sections were then deparaffinized, rehydrated, rinsed, and placed in PBS before harrys hematoxylin staining (Diapath, diluted 1/3), washed with PBS and stained with eosin 1% (Sigma-Aldrich). Slide were washed with water dehydrated and mounted with DPX mounting medium (Sigma-Aldrich). For the study of eosinophils infiltration, slides were stained in addition with saffron and analysed by a board-certified veterinary pathologist.

## Flow cytometry analysis

Whole blood was collected on EDTA, then transferred into BD vacutainer CPT tubes (after removal of anticoagulant solution from CPT tubes) and spun at 2500 g for 20 min. The PBMCs were collected and one-tenth were used to isolate RNA. The remaining cells were surface stained on ice using three different panels, including Panel A: CD150 BV-421 (clone A12, BD), CD8 AF 647 (clone RPA-T8), CD20 APC-H7 (clone 2H7, BD), CD3 V500 (clone SP34-2, BD), CD14 Pe-cy7 (clone M5E2, BD); Panel B: CD150 BV421, CD3 V500, CCR7 Pe-Cy7 (clone G043H7, Biolegend), CD8 AF647, CD45RA APC-H7 (clone 5H9, BD); and Panel C: CD150 BV421, IgD BV510 (clone IA6-2, BD), CD38 Pe-Cy7 (clone HB7, BD), CD27 AF647 (clone O323, Biolegend), CD20 APC-H7, using 4 μl of each Ab per sample. Cells were acquired on a MACS-Quant®10 flow cytometer (Miltenyi) and analyzed by FlowJo10.8 (Becton Dickinson) and Kaluza 2.1 (Beckman Coulter). Gating strategy is presented in supplementary fig. S8.

## Statistical analysis

We used the 1-way and 2-way ANOVA analysis, Mantel Cox and Mann-Whitney tests for statistical analyses of results of the fusion test and animal survival and non-linear regression for the calculation of the serum neutralization titer. We considered p-values of 0.05 or below (two-tailed tests) to be statistically significant. Statistical analyses were performed using GraphPad Prism 8.3 software.

**Reporting summary**

Further information on research design is available in the Nature Research Reporting Summary linked to this article.

## Data availability

Source data of figs.: 1c, 2b, d, 3c, d, 4a, b, 6d, 7a, b, 8a, c, s3, s4, s6 and s7 are provided with this paper. Other raw data files (flow cytometry data and confocal images) are available on request to the corresponding author. Reference genome of measles virus used in the study, MeV-IC323-eGFP, is available at Genbank with accession number: LC420351.1. Sequencing reads are deposited in NCBI BioProject PRJNA828179. Source data are provided with this paper.

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

## Acknowledgements

We thank the animal experimentation team of Tours University for the realization of the animal experiments. We are grateful to Dr Cyrille Debard, (Biovelys, VetagroSup), Dr Guillaume Noel (Biovivo, VetagroSup) for helpful veterinary advices, G. Gourru-Lesimple for mycoplasma tests, Adam Sidotmane for the technical assistance in characterization of nebulizers, Dr Emmanuelle Guillot-Combe for stimulative discussions on the utilization of nebulization approach to prevent airborne viral infections and all the members of the group Immunobiology of viral infection at CIRI for the help during the realisation of this study. We acknowledge the Servier Medical Art (smart.servier.com) for providing the images used for the schemas presented in the article and the contribution of the SFR Biosciences (UMS3444/CNRS, US8/Inserm, ENS de Lyon, UCBL) facility Lymic-Platim-Microscopy and AniRA PBES. The study was supported by Region ARA (Pack Ambition Recherche, project AerVirStop-BH), by LABEX ECOFECT (ANR-11-LABX-0048-BH) of Lyon University, within the program "Investissements d'Avenir" (ANR-11-IDEX-0007-BH) operated by the French National Research Agency (ANR), and by ANR 16-ASMA-0008-01 (BH) and by NIH NS09126, NS105699 and AI159085(MP).

## Author contributions

Conceptualization: B.H., O.R., L.V., M.I., A.M., M.P., C.M.; Methodology: O.R., C.D., C.G., M.I., S.L.G., L.V., J.M., C.M., A.L.G., C.A.A.; Investigation: O.R., L.V., J.M., E.B., M.I., C.D., C.G., F.T.B., S.L.G., C.M., L.L., M.F., Y.Z.,

D.L.P., G.C., G.R., A.A.; Funding acquisition: B.H., M.P.; Project administration: B.H., M.P.; Supervision: B.H., O.R., C.D., M.I., C.M.; Writing, original draft: O.R., M.I., B.H.; Writing, review and editing: O.R., B.H., M.I., M.P., A.M.

## Competing interests

S. Le Guellec is employed by DTF Medical (Saint Etienne, France) and L. Vecellio was employed by DTF Medical from 2001 to 2018 and by Nemera (La Verpilliere, France) from 2018 to 2020. The remaining authors declare no competing interests.

## Additional information

[1]CIRI, Centre International de Recherche en Infectiologie, INSERM U1111, CNRS, UMR5308, Univ Lyon, Université Claude Bernard Lyon 1, École Normale Supérieure de Lyon, 21 Avenue Tony Garnier, 69007 Lyon, France. [2]DTF-Aerodrug, R&D aerosolltherapy department of DTF medical (Saint Etienne, France), Faculté de médecine, Université de Tours, 37032 Tours, France. [3]Université de Tours, Institut national de recherche pour l'agriculture, l'alimentation et l'environnement (INRAe), UMR1282, Infectiologie et santé publique (ISP), Tours, France. [4]PST-A, Université de Tours, Tours, France. [5]Center for Host-Pathogen Interaction, Department of Pediatrics, Columbia University Vagelos College of Physicians and Surgeons, New York, NY, USA. [6]Laboratory of Infection and Virology, Beijing Children's Hospital, Capital Medical University, National Center for Children's Health, Beijing 100045, China. [7]INSERM, Research Center for Respiratory Diseases, CEPR U1100, Université de Tours, 37032 Tours, France. [8]Department of Laboratory Medicine and Pathology, University of Washington Medical Center, Seattle, WA, USA. [9]Robert Frederick Smith School of Chemical and Biomolecular Engineering, Cornell University, Ithaca, NY, USA. [10]UMR703, PAnTher APEX, INRAE/Oniris, Nantes, France. [11]Department of Microbiology and Immunology, Columbia University Vagelos College of Physicians & Surgeons, New York, NY, USA. [12]Department of Physiology & Cellular Biophysics, Columbia University Vagelos College of Physicians & Surgeons, New York, NY, USA. [13]Department of Experimental Medicine, University of Studies of Campania 'Luigi Vanvitelli', Naples, Italy. ✉e-mail: branka.horvat@inserm.fr

