## [Peer Review File · Nature Communications]

REVIEWER COMMENTS

Reviewer #1 (Remarks to the Author):

In this manuscript, Reynard et al. test the efficacy of a peptide-based fusion inhibitor against measles virus (MeV) in a non-human primate model, cynomolgus macaques. To block virus from invading the host through the respiratory tract, they develop an aerosol-delivery system for the inhibitor, and evaluate both the distribution of the inhibitor in the lungs of treated animals, as well as its efficacy against wild type MeV infection. The inhibitor has previously been developed and is based on the HRC-domain of the MeV fusion protein. It interacts with F and prevents its ability to undergo a conformational change required for membrane fusion. They show that the drug is efficiently deposited in lung tissue after aerosolization, and they do not detect notable immune or adverse reactions in the animals. The inhibitor showed excellent ability to prevent MeV infection, when animals were treated three times in a time frame from 1 day prior to infection to one day post infection. Notably, one treated animal that was co-housed with a mock-treated animal developed MeV infection at a later time point indicating that virus was transmitted from the control animal to the treated animal at a time point after drug-clearance. Importantly, animals that were protected from MeV infection by the drug did not develop any MeV-specific immunity, indicating that the drug indeed completely prevented infection. The results indicate that the fusion inhibitory drug may be an excellent tool to prevent MeV transmission in an outbreak scenario.

The manuscript is very well written and the conclusions are supported by the data. I only have few comments that should be addressed by the authors:

1. In Fig. 7d, the legend to the diagrams indicates that animal P3 developed MeV infection at a later stage, and not, as described in the text animal P2. However, P3 was co-housed with animal P1, which was also treated with HRC4-peptide. The authors need to verify their data sets and either correct the graphs or the statements on the potential transmission from control to peptide-treated animal at a later stage of infection.
2. In Fig. 4, the authors evaluate blood parameters of cynomolgus macaques after peptide treatment. While most parameters showed no striking alterations compared to control-treated animals, it seems like eosinophils and basophils are both consistently low in peptide-treated animals, but not in control-treated animals. Low blood levels of these two cell types can indicate ongoing allergic reaction, and these cells may have infiltrated the lung tissue, where HRC4-peptide was deposited. The authors should evaluate in histology, whether lungs exhibit enhanced eosinophil and/or basophil infiltration in the peptide-treated animals.

3. In line with this, the authors should also investigate whether HRC4-specific IgE is present in the serum of peptide-treated animals. In Fig. 3d, they measure IgG, IgA, and IgM, but not IgE. If IgE is detectable, this may indicate that there is a potential for allergic reaction towards HRC4, and this would affect the safety profile of the drug.

4. Since protected animals do not develop MeV immunity, it would be very important to elaborate on the clinical use of HRC4 in the discussion. While the drug may be very effective in preventing infection, its use will not necessarily aid the eradication of MeV, as treated individuals would remain vulnerable to MeV infection. Do the authors think, treatment of high risk individuals could be accompanied with parallel immunization with the MMR vaccine, or is it possible that the drug may also interfere with vaccine efficacy?

Minor comments:

- In line 230, the formula contains the term "57.8,106". This seems to be a typo, as I cannot make any sense out of it.

- Fig. 8b: There seems to be no data point for animals C1 and C2 on day 28. (Maybe also in fig 8a, but it is hard to say.) Why?

Reviewer #2 (Remarks to the Author):

In this manuscript, Reynard et al. report that aerosolized lipopeptide fusion inhibitors, which were derived from heptad-repeat regions of the measles virus (MeV) fusion protein, block respiratory MeV infection in a NHP model, the cynomolgus macaque. They used a custom-designed mesh nebulizer to deliver the peptides to the respiratory tract and demonstrated the absence of adverse effects and lung pathology, and the nebulized peptides efficiently prevented MeV infection, resulting in the complete absence of MeV RNA, MeV-infected cells, and MeV-specific humoral responses in treated animals. This is an important proof-of-concept study to show the effectiveness of nebulized fusion peptide inhibitors as antivirals.

Here are a few issues which need to be addressed:

1. If this treatment is meant for young children, how can one ensure that children adequately inhale the peptides?

2. The authors claim that this treatment can serve as an alternative to vaccination; however, they also reported that one of the treated NHPs likely became infected from their mock-treated cage-mate. They hypothesize that this is because they had stopped nebulization of the peptide, if so how can they argue that this treatment offers protection similar to the current MMR vaccine if the treated NHP still developed infection? (MMR vaccine is ~93% efficient against measles according to the CDC) - Lines 346-355.

3. If this treatment is meant to be used as a co-treatment to the current MMR vaccine, will they test this treatment in vaccinated subjects? - Lines 50-52

4. Statistical analysis/significance should be added to the graphs in all figures.

5. In figure 7b, C2 has a much lower n value than the other C1, C3 or P2, is this data still significant?

6. In figure 1a, can authors please simplify the arrows to make a more concise figure?

NCOMMS-22-21155: "Nebulized fusion inhibitory peptide protects cynomolgus macaques from measles virus infection".

Answers to reviewers

Reviewer 1:

The manuscript is very well written and the conclusions are supported by the data. I only have few comments that should be addressed by the authors:

1. In Fig. 7d, the legend to the diagrams indicates that animal P3 developed MeV infection at a later stage, and not, as described in the text animal P2. However, P3 was co-housed with animal P1, which was also treated with HRC4-peptide. The authors need to verify their data sets and either correct the graphs or the statements on the potential transmission from control to peptide-treated animal at a later stage of infection.

We thank the reviewer for the positive evaluation of our manuscript and the constructive remarks. The exchange of the numbers of animals was indeed made in the Figure 6d; we apologize for that omission which has been corrected in the revised manuscript.

2. In Fig. 4, the authors evaluate blood parameters of cynomolgus macaques after peptide treatment. While most parameters showed no striking alterations compared to control-treated animals, it seems like eosinophils and basophils are both consistently low in peptide-treated animals, but not in control-treated animals. Low blood levels of these two cell types can indicate ongoing allergic reaction, and these cells may have infiltrated the lung tissue, where HRC4-peptide was deposited. The authors should evaluate in histology, whether lungs exhibit enhanced eosinophil and/or basophil infiltration in the peptide-treated animals.

We have re-examined all the data linked to hematological parameters presented in the figure 4. The mean number of eosinophils in both control and peptide treated group is equivalent, with lower variation in peptide treated group. The possible lung infiltration of eosinophils was extensively analyzed by a board-certified pathologist, as described in the Methods section and obtained results added in supplementary Fig. S4c and commented in lines 182-188 of the revised manuscript. Although at low level, an infiltration of eosinophils was found in all animals that developed infection but not in the two animals fully protected by HRC4, excluding thus the effect of peptide on the eosinophil infiltration and supporting the link between measles infection and eosinophil migration into the lungs, as reported in previous studies (ref N°40, Polack *et al*). In support to the absence of allergic reactions in lungs, the histological analysis of lung sections did not present in the airways any signs of hyperplasia of mucus cells and smooth muscles, characteristic for allergic reactions of the respiratory tract. Finally, surprisingly high level of basophils in the control group resulted from the error in the graphical presentation in the Fig. 4b and has been corrected in the revised manuscript.

3. In line with this, the authors should also investigate whether HRC4-specific IgE is present in the serum of peptide-treated animals. In Fig. 3d, they measure IgG, IgA, and IgM, but not IgE. If IgE is detectable, this may indicate that there is a potential for allergic reaction towards HRC4, and this would affect the safety profile of the drug.

We have analyzed the presence of total and peptide-specific IgE in the serum of macaques using ELISA approach and presented the data in supplementary Fig. S4a and b, with comments on lines 195-197. We did not detect any HRC4-specific IgE, nor increase of total IgE in control and peptide-treated animals during the 4 weeks after treatment, excluding thus IgE-mediated allergic reaction to the nebulized peptide.

4. Since protected animals do not develop MeV immunity, it would be very important to elaborate on the clinical use of HRC4 in the discussion. While the drug may be very effective in preventing infection, its use will not necessarily aid the eradication of MeV, as treated individuals would remain vulnerable to MeV infection. Do the authors think, treatment of high risk individuals could be accompanied with parallel immunization with the MMR vaccine, or is it possible that the drug may also interfere with vaccine efficacy?

We completely agree with the reviewer that peptides cannot substitute the vaccination in the eradication of measles. As mentioned in the discussion (lines 400-402), HRC4 nebulization holds potential for protecting immunocompromised people who rely on herd immunity and cannot receive the current live MeV vaccine. To further elaborate the clinical use of HRC4, we have added in discussion lines 402-406: "In addition, certain temporal conditions (blood transfusion, transplantation, pregnancy, tuberculosis, etc) may require a postponement of measles vaccination⁶⁴, presenting the situation where HRC4 nebulization may provide a solution until vaccination is again possible and could even be continued after vaccination until appearance of protective immunity."

Minor comments:

- In line 230, the formula contains the term "57.8,106". This seems to be a typo, as I cannot make any sense out of it.

We apologize for that error; the typos have been corrected: "5.78 x 10⁷"

- Fig. 8b: There seems to be no data point for animals C1 and C2 on day 28. (Maybe also in fig 8a, but it is hard to say.) Why?

We thank the reviewer for pointing out this omission, which has been corrected in the revised manuscript.

Reviewer 2:

This is an important proof-of-concept study to show the effectiveness of nebulized fusion peptide inhibitors as antivirals.

We thank the reviewer for valuing the contribution of this work.

Here are a few issues which need to be addressed:

1. If this treatment is meant for young children, how can one ensure that children adequately inhale the peptides?

The HRC4 administration by nebulization can be applied to both adults and children in case where vaccination is not recommended, not effective or not currently implemented. Several approved drugs are already delivered by nebulization to young children, notably for the treatment of acute bronchiolitis and pulmonary infections (References: PMID: 35509393, PMID: 35383741, PMID: 34059219). In contrast to metered-dose powder

inhalation devices used notably for the treatment of asthma (for the distribution of bronchodilators and corticoids), where the short nebulization time may lead to the administration of variable dose of drug (Cho-Kang&Grant, PMID: 1092420), possible mid-length nebulization time of HRC4 peptide in young children below 5 years, with 300 to 500 breathes occurring during administration period of 10 min (Fleming et al, PMID: 21411136), should thus allow minimal loss of drug during administration, optimizing its distribution in the respiratory tract. Finally, as shown by Asgharian B et al, PMID: 23121298, comparison of aerosol delivery between children and primates tends to be more favorable in children, with higher pulmonary deposition than in rhesus monkey. Altogether, these data strongly support the possibility to use peptide nebulization in young children.

2. The authors claim that this treatment can serve as an alternative to vaccination; however, they also reported that one of the treated NHPs likely became infected from their mock-treated cage-mate. They hypothesize that this is because they had stopped nebulization of the peptide, if so how can they argue that this treatment offers protection similar to the current MMR vaccine if the treated NHP still developed infection? (MMR vaccine is ~93% efficient against measles according to the CDC) - Lines 346-355.

We completely agree with the reviewer that peptides cannot substitute for the vaccination and could give only **short-term** protection against infection. We indeed consider that vaccination is the key element to control infectious disease and we have modified lines 402-406, to underline that such treatment could not present a life-long treatment but may be used in specific cases as a temporary solution to protect immunosuppressed patients who could not be vaccinated, or in the cases of infection when vaccine is not available. Our results suggest that HRC4 has a limited half-life after deposition in lungs and most likely three days after the last nebulization the concentration of the lipopeptide in lungs is not sufficient anymore for the efficient protection.

3. If this treatment is meant to be used as a co-treatment to the current MMR vaccine, will they test this treatment in vaccinated subjects? - Lines 50-52

This treatment is not expected to be used as co-treatment with MMR vaccine except in case of vaccine failure; the principal targeted population is expected to be nonvaccinated people with the risk of exposure to measles virus and immunocompromised individuals who could not be vaccinated, which has been now additionally specified in the abstract of the revised manuscript.

4. Statistical analysis/significance should be added to the graphs in all figures.

Statistical analysis and significance have been added in figures of the revised manuscript. Figure 2c and d, which initially showed a representative experiment, have been completed and present now a sum of three independent experiments, analyzed using a Mann-Whitney test. In most of the other figures the statistical analysis has been either already included or the curves present individual animals and obtained results correspond to either development of the response or to its complete absence, without necessity for an additional statistical analysis.

5. In figure 7b, C2 has a much lower n value than the other C1, C3 or P2, is this data still significant?

We agree with the reviewer that the C2 animal had much lower n value than other macaques, which is probably due to the lower level of infection in that animal (as seen in the Fig 6d). To avoid any confusion, we thus removed the C2 graph plot from the figure 7b in the revised manuscript.

6. In figure 1a, can authors please simplify the arrows to make a more concise figure?

The figure 1a has been modified in accord to the reviewer's suggestion.

REVIEWERS' COMMENTS

Reviewer #1 (Remarks to the Author):

Thank you very much for addressing all my comments. I have no additional requests.